# The Impact of the COVID-19 Pandemic on Adherence to Endocrine Therapy for Breast Cancer in Catalonia (Spain)

**DOI:** 10.3390/cancers16020426

**Published:** 2024-01-19

**Authors:** Aurea Navarro-Sabate, Rebeca Font, Fina Martínez-Soler, Judit Solà, Avelina Tortosa, Josepa Ribes, Llúcia Benito-Aracil, Josep Alfons Espinas, Josep Maria Borras

**Affiliations:** 1Fundamental Care and Clinical Nursing Department, Nursing Faculty, University of Barcelona, 08007 Barcelona, Spain; aureanavarro@ub.edu (A.N.-S.); finamartinez@ub.edu (F.M.-S.); atortosa@ub.edu (A.T.); lbenito@ub.edu (L.B.-A.); 2Catalan Cancer Plan, Department of Health, 08908 Barcelona, Spain; rfont@iconcologia.net (R.F.); j.ribes@ub.edu (J.R.); ja.espinas@iconcologia.net (J.A.E.); 3Bellvitge Biomedical Research Institute (IDIBELL), Hospitalet, 08907 Barcelona, Spain; 4Clinical Sciences Department, University of Barcelona, 08907 Barcelona, Spain

**Keywords:** adherence, breast cancer, COVID-19, endocrine therapy, real-world data

## Abstract

**Simple Summary:**

Oral endocrine therapy is a key component of breast cancer treatment in hormone-positive patients, the most frequent subtype of this cancer. Adherence to this treatment is a factor related to an increased probability of relapse and death. The COVID-19 pandemic has had a disruptive impact on the use of cancer services, delaying diagnosis and stopping screening during the peak of the pandemic. We assessed the impact of the COVID-19 on the adherence to oral endocrine therapy in Catalonia (Spain) using the data collected in the clinical records of the public health care system. We have found that the first wave of the COVID-19 (between March and September 2020) was associated with a modest impact on treatment adherence.

**Abstract:**

**Purpose**. To assess the impact of the COVID-19 pandemic on adherence to oral endocrine therapy in patients diagnosed with breast cancer in the public healthcare system in Catalonia (Spain). **Methods**. Retrospective cohort study in patients starting endocrine therapy from 2017 to 2021. Adherence was measured during the first year of treatment, and the impact of the pandemic was calculated according to the calendar year and whether the first year of treatment included the peak period of the pandemic in our setting (March–September 2020). Analyses were performed using a chi-square test and multivariable logistic regression, with results stratified by year, age group, and drug type. **Results**. Mean overall adherence during the first year of treatment was 89.6% from 2017 to 2021. In contrast, the patients who started treatment in 2019 and 2020 and whose treatment included the peak pandemic period presented an adherence of 87.0% and 86.5%, respectively. Young age and tamoxifen or combination therapy were predictors of low adherence. An increase in neoadjuvant therapy was also observed in 2020. **Conclusions**. The COVID-19 pandemic had only a modest impact on adherence to endocrine therapy (≈3%), despite the enormous disruptions for patients, the healthcare system in general, and cancer care in particular that were occurring in that period.

## 1. Introduction

Breast cancer is the most common cancer worldwide, with approximately 2.3 million new cases in 2020, surpassing lung cancer as the most frequently diagnosed [1]. About 27,700 new cases of breast cancer were diagnosed in Spain in 2015, including 5500 in Catalonia [2,3].

Endocrine therapy with tamoxifen and/or aromatase inhibitors is a mainstay of adjuvant treatment for hormone receptor-positive breast cancer, the most common molecular subtype [4,5]. The importance of these drugs is reflected in clinical guidelines [6,7,8] and supported by a large body of evidence showing their benefits for reducing the risk of recurrence and subsequent mortality [9]. The recommended duration of treatment is at least 5 years, although some authors argue that the long-term risk of recurrence justifies treatment for up to 10 years, as long as the risk–benefit balance is favorable [10,11,12]. The benefits of endocrine therapy hinge on patients’ consistent adherence over the course of the treatment period. Adherence of less than 80% to the prescribed medication, as assessed by continuous measurement of prescription refill in the community pharmacy, has been associated with an increased risk of recurrence and all-cause death [13,14,15,16]. Therefore, analyzing adherence is relevant when evaluating the prognosis of patients undergoing active treatment for breast cancer [11].

COVID-19 declared a public health emergency by the World Health Organization (WHO) in January 2020 [17], caused enormous disruptions to patients and healthcare systems worldwide, as probably no other emergency had before. Almost immediately, it was necessary to reallocate a large portion of the available health services resources to the care of patients with COVID-19, which had a notable impact on the management of other diseases. In Catalonia (Spain), the overall number of cancer diagnoses fell by 19% during the first three months of the pandemic. In breast cancer, this reduction was estimated at 45% between March and July 2020, coinciding with the first wave of the pandemic. This decrease in new diagnoses of breast cancer was especially sharp (52%) in women aged 50 to 69 years, the target group of our country’s population-based screening, which was suspended during this period [18]. A literature review of the impact of COVID-19 on screening and diagnosis reported similar patterns elsewhere, with reductions of more than 25% in breast cancer diagnoses. In addition, review authors found higher proportions of women diagnosed with tumors at more advanced stages [19]. Finally, the pandemic reduced face-to-face visits in hospitals and hospitalizations for breast cancer [19,20,21].

In this context and given the lack of evidence of the relationship between COVID-19 and adherence to breast cancer treatment, the aim of this study was to assess the impact of the peak pandemic period (March to September 2020) on adherence to endocrine therapy in the Catalan population treated in the public healthcare system, relative to the preceding period (2017 to 2019) and its post-pandemic evolution (to 2022).

## 2. Material and Methods

This retrospective cohort study included 18,131 patients with breast cancer who initiated oral endocrine therapy from 2017 to 2021, with follow-up to December 2022.

Data were drawn from the pharmacy database of the Catalan public system, which stores information on the prescription and dispensation of the drugs used for oral endocrine therapy (tamoxifen and aromatase inhibitors). Prescription of these oral therapies is always performed by a hospital specialist, but the drugs should be refilled in a community pharmacy. No disruptions were observed in the provision of these drugs during the pandemic. The date of dispensation, the type of drug administered (tamoxifen, aromatase inhibitors), and the number of pills dispensed were recorded for each patient. The date of the first dispensation at the pharmacy was considered the start date of the hormonal treatment. We did not know the date of diagnosis, only the date of first refilling at community pharmacy. Women who started endocrine therapy in 2022 were excluded, as it was not possible to calculate adherence at one year of treatment (n = 3409), since follow-up ended in December 2022 and the full year was not available.

The study cohort was linked to the hospital discharge minimum basic dataset in Catalonia to identify women who had been admitted to a public hospital with a primary diagnosis of breast cancer (International Classification of Diseases, 10th revision (ICD-10): C50, D05.1, D05.7 and D05.9). Date of surgery was collected from the hospital discharge minimum dataset. In this database, stage at diagnosis was not included and we were not able to include this variable in the analysis. For each woman in this study population (n = 18,131), age at first admission was recorded. Patients treated in private centers (n = 5542) were excluded, as their data were not available in the minimum basic dataset. The percentage of patients receiving neoadjuvant endocrine therapy was calculated as the proportion of cases with surgery after endocrine treatment. Finally, the cohort was linked to the Catalan Health Department’s Central Insurance Registry to determine each patient’s vital status and the date of death to define the last time the drug was used, if applicable.

The primary outcome was adherence to oral endocrine therapy during the first year of treatment, with adherence defined as “the extent to which a patient performs according to the prescribed interval and dose of a dosing regimen” [22]. A change of drug was considered a continuation of treatment. Adherence was considered satisfactory if the coverage of the prescribed medication was at least 80% [16]. Adherence was first compared according to the year of treatment initiation, and then, more precisely, according to whether the women’s first year of treatment concluded before March 2020 (pre-pandemic), overlapped with the peak pandemic period (March–September 2020), or began after October 2020 and followed until December 2022 (post-pandemic). Other explanatory variables were age at the start of endocrine therapy (categorical: ≤49 years, 50–69 years, and ≥70 years) and type of oral hormonal treatment (aromatase inhibitors, tamoxifen, or combined therapy (switch during treatment)).

We performed a descriptive statistical analysis of the study variables as well as univariable and multivariable logistic regressions, using adherence as the outcome. Results are expressed as odds ratios (OR) with their 95% confidence intervals (CI). The multivariable model was adjusted for year, age, and type of treatment. A test for trend was performed to assess the administration of neoadjuvant oral endocrine therapy. All analyses were performed using the SPSS software package (Armonk, NY, USA; version 21). The study was approved by the ethics committee for drug research at Hospital Universitari de Bellvitge on 25 October 2022 (reference **EOM028/22**).

## 3. Results

From 2017 to 2021, 21,540 women were diagnosed with breast cancer in the public healthcare system and treated with oral endocrine therapy; of these, 18,131 had adherence data for a full year of follow-up. Over the complete study period (from 2017 to the end of 2022), 1184 women died (Figure 1).

Table 1 shows the breast cancer patients according to the year they started endocrine therapy as well as their adherence status, overall and by age group and drug type. The mean number of women treated with hormone therapy annually in Catalonia before the pandemic (2017–2019) was 3614.7. In 2020, this figure dropped by 5.4% before rebounding in 2021, with increases of 7%. A notable reduction in adherence to endocrine therapy was observed in women starting treatment in 2019 and 2020 (many of whose first year of treatment overlapped with the pandemic period) relative to those who began in 2017 and 2018. A complementary analysis, grouping women according to whether their first year of treatment included a month of the pandemic period, showed lower adherence rates in those women treated during the pandemic period compared to those treated before or after (pre-pandemic: 89.8%; pandemic: 86.6% and post-pandemic: 88.6%; Table 2).

Regarding the relationship between adherence during the first year of treatment and age at first dispensation of medication (Table 1), adherence rates surpassed 88% in the largest age group (50–69 years), regardless of the year treatment began, with similar results observed in the >70 years age group. In contrast, women aged 50 years or under presented lower adherence rates—at least 4 percentage points below in comparison with the rest of the age groups in all the years analyzed. When analyzed according to pandemic periods, this pattern held true for all age groups except those over 70 years of age (Table 2).

Regarding drug type, adherence in the first year of oral endocrine therapy was consistently higher in patients treated with aromatase inhibitors and consistently lower in those receiving combined therapy (Table 1). The reduced adherence rates observed in 2020 tended to recover in 2021 in women receiving tamoxifen or combined therapy. Among all treatment groups, adherence rates decreased in women treated during the pandemic period and recovered post-pandemic (Table 2).

The multivariable analysis showed that treatment starting in 2019 and 2020 was associated with lower adherence, and the difference with respect to the baseline year of 2017 was significant (Table 3). Age over 50 years was independently associated with greater adherence, after adjusting for year and drug type. Moreover, women taking aromatase inhibitors were somewhat more adherent, and those receiving a combination of drugs significantly less so (Table 3).

Figure 2 shows the significant increase in the percentage of patients receiving neoadjuvant oral endocrine therapy from 2017 (3.8%) to 2020 (5.8%) before it dropped again in 2021 (4.7%; *p* < 0.003 for trend).

## 4. Discussion

Our analysis of women with breast cancer starting oral endocrine therapy from 2017 to 2021 showed that the first wave of the COVID-19 pandemic (between March and September 2020) had only a modest impact on treatment adherence. However, there was a notable reduction in the absolute number of patients who started this treatment in 2020 (around 250 fewer new cases, Table 1), although 2021 saw a full recovery to pre-pandemic levels. In addition, a change in the therapeutic pattern took place during the pandemic, with the probability of receiving neoadjuvant treatment increasing in 2020.

We started by analyzing adherence according to the first year of treatment with endocrine therapy to allow an interannual comparison over the three years before and two years after 2020, the peak year of the pandemic. The results show a slight decrease in adherence rates in women starting treatment in 2019 and 2020, of around 3% compared to 2017–2018. The decrease is probably attributable to the pandemic given that the reduction observed for the 2019 cohort includes their treatment in 2020. This conclusion is supported by the analysis specifically comparing patients based on whether their first year of oral endocrine therapy included the peak months of the pandemic (March–September 2020 in our country); the data confirm a modest impact on adherence rates, with a reduction of approximately 3% (Table 2). In any case, given the recovery in adherence in 2021, this impact seems to have been temporary and concentrated in the period of greatest disruption to the health system. In women under 70 years of age, the recovery in 2021 was very notable and statistically significant, while in older women, there was no noticeable impact at all.

The greatest impact was observed in the number of patients who started oral endocrine therapy in 2020 in the first place, due to a sharp reduction in diagnoses. These data are consistent with observed diagnostic delays for practically all tumors, including breast cancer, both in our setting [18] and internationally [23,24]. The observed 5.9% reduction in diagnoses in 2020 can be largely explained by the suspension of screening programs during the four-month lockdown in the first phase of the pandemic. Accordingly, there was a substantial absolute decrease in diagnoses in women aged 50 to 69 years—the target age group for population-based screening in our country. After this period, population screening rates progressively recovered, and participation returned to its normal pre-pandemic values [25]. Another aspect that could have influenced the reduction in the number of diagnoses in this initial period of the pandemic was the fear of presenting to the health services with symptomatic disease [26,27]. The pandemic also entailed an increase in neoadjuvant treatment, which was probably used to extend the interval between diagnosis and surgery; this is one of the strategies described by professionals to deal with the circumstances characterizing the early pandemic period [28,29]. Even though oncological surgery was preserved as far as possible, public health restrictions undoubtedly limited access to hospitals during the first pandemic wave (March–July 2020). Afterwards, the impact on hospital occupancy was notably more limited [18], which could explain the return to the usual rates of neoadjuvant therapy.

Non-adherence in our country has typically been associated with age and type of endocrine therapy. Adverse effects seem play a more limited role than suggested by data from Nordic countries or the USA [15,16]. In our study, the two most relevant factors associated with non-adherence were age under 50 years and starting treatment with tamoxifen. The cases of combined therapy in the first year of treatment may be explained by therapeutic switching due to adverse effects, which would in turn explain the significantly lower adherence in this group of patients compared to those on individual therapies and the notable decrease in the peak period of the pandemic (Table 1 and Table 2).

Our study has both limitations and strengths that should be considered when interpreting data. In the first place, we used real-world data, which means that variables like stage at diagnosis were unavailable, a factor clearly related to adherence and the multidisciplinary treatment strategy [15]. Likewise, it was not possible to differentiate between endocrine therapy in tumors with an early versus advanced stage. However, our data do allow a quick analysis of diagnosis and adherence in light of the disruption in health services caused by COVID-19, especially in the first wave and until the introduction of vaccination in 2021. In the present study, we mainly aimed to assess the impact in 2020, as hospital activity basically recovered in 2021 [18]. The fact that age and type of therapy showed a similar magnitude of association as previous studies using population-based or hospital data reinforces the validity of our approach based on real-world data. Also of note, the analysis was carried out year by year due to the difficulty of calculating the prescription refill in quarterly periods. Although such an approach would increase the granularity and precision of the analysis, the risk of overlapping prescriptions and the variability of refill timing were factors that prompted us to extend the period of analysis up to one year. Therefore, to compare the pandemic period with preceding and subsequent periods, we focused on adherence in the first year after the therapeutic indication. Although this measure is associated with 5-year adherence, the results in our cohort need to be confirmed in the coming years. Another consideration is the use of prescription refill data as the adherence measure, with the assumption that dispensation is a good proxy for consumption [30]. This study is based on patients who effectively initiated treatment refilling the drug in a community pharmacy, this implies that the lack of adherence due to not refilling any drugs to start the treatment was not assessed in this study. It should be mentioned that this study is focused on evaluating the potential impact of a limited number of factors associated with adherence, like age and type of therapy; but there are other very relevant factors not considered due to the real-world data approach used, such as psychological or barriers to access [30]. Finally, we focused on patients treated with both surgery and endocrine therapy in the public system, excluding those operated on in the private system even if they were later prescribed endocrine therapy through public health care services, a situation permitted in our setting.

## 5. Conclusions

In conclusion, we did not observe a major impact of the COVID-19 pandemic on adherence in women with breast cancer starting treatment with endocrine therapy, despite the great impact caused by the suspension of screening programs; the disruptions to medical care, hospital visits, and primary care; and by the lockdown itself (especially strict from March to July 2020, with progressive relaxation over the second half of 2020). The factors associated with low adherence were similar to those observed in the pre-pandemic period, such as young age, combination therapy, or treatment with tamoxifen. Finally, the pandemic led to a temporary increase in the use of neoadjuvant therapy as a means of delaying surgery in 2020. It would be of interest to evaluate the 5-year persistence and adherence of these annual pre-, post-, and peak pandemic cohorts in the future in order to assess the temporal impact and its possible influence on clinical outcomes, if any. The speed with which such impacts can be assessed using real-world data is an advantage when estimating the consequences of disruptions to health services for cancer treatment.

## Figures and Tables

**Figure 1 cancers-16-00426-f001:**
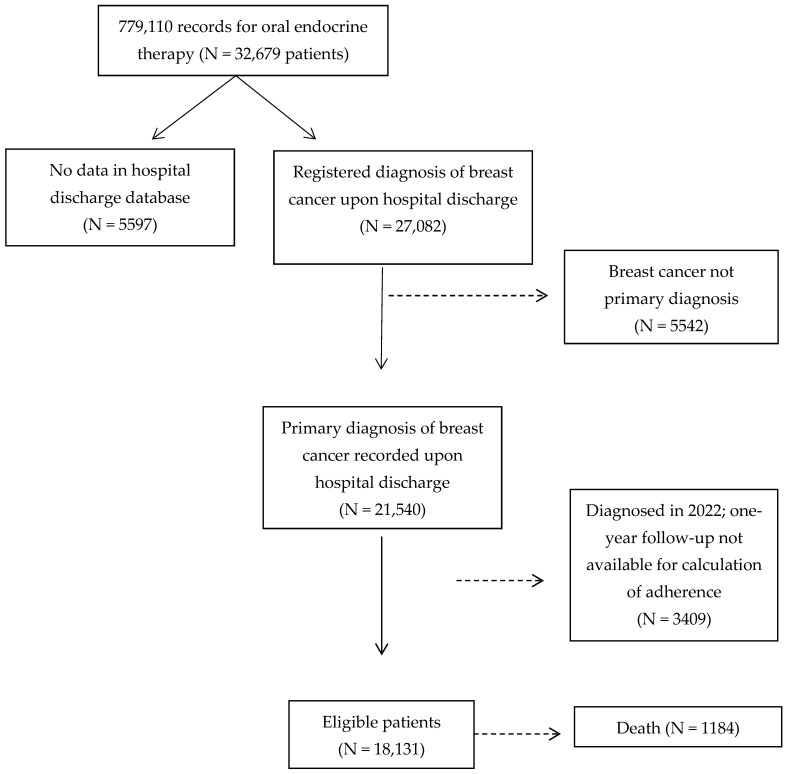
Flow chart of patients diagnosed with breast cancer through the public healthcare System in Catalonia (Spain), 2017 to 2021.

**Figure 2 cancers-16-00426-f002:**
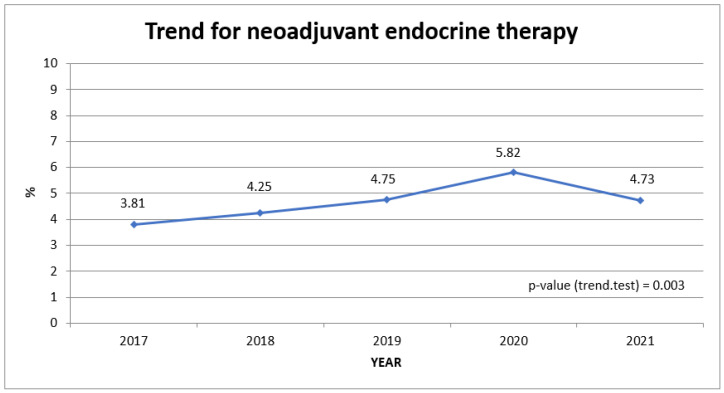
Proportion of patients receiving neoadjuvant endocrine therapy over study period.

**Table 1 cancers-16-00426-t001:** Adherence to oral endocrine therapy at one year from treatment initiation, 2017 to 2021 (followed until December 2022).

	Total Study Period	Year of Treatment Initiation					*p* *
		2017	2018	2019	2020	2021	
**Overall adherence ^†^, n/N (%)**	16,059/18,131 (89.6)	3260/3613 (90.2)	3195/3555 (89.9)	3198/3676 (87.0)	2960/3420 (86.5)	3446/3867 (89.1)	0.002
**N total patients in each age group (adherence rate)**							
<50 years	3942 (82.4)	794 (85.6)	760 (83.7)	833 (79.7)	758 (78.5)	797 (84.7)	0.108
50–69 years	9376 (90.6)	1929 (92.4)	1866 (92.9)	1843 (88.4)	1744 (88.5)	1994 (90.6)	<0.001
≥70 years	4813 (89.6)	890 (89.6)	929 (88.8)	1000 (90.4)	918 (89.5)	1076 (89.6)	0.801
**N total patients in each treatment group (adherence rate)**							
Tamoxifen	4618 (86.8)	1040 (89.8)	920 (89.5)	976 (85.2)	807 (83.8)	875 (85.0)	<0.001
Aromatase inhibitors	12,455 (90.2)	2349 (91.5)	2432 (90.7)	2509 (88.4)	2389 (88.8)	2776 (91.4)	0.39
Combined treatment	1058 (77.4)	224 (79.0)	203 (82.3)	191 (77.0)	224 (72.3)	216 (76.9)	0.11

* Chi-squared test for trend. ^†^ Adherence defined as 80% coverage of prescribed dose.

**Table 2 cancers-16-00426-t002:** Adherence to oral endocrine therapy at one year from start of treatment during different pandemic periods, by age and type of treatment.

	Pre-Pandemic ^†^ n/N (%)	Pandemic ^†^ n/N (%)	Post-Pandemic ^†^ n/N (%)	*p* ^‡^
**Overall adherence ***	7278/8106 (89.8)	4434/5118 (86.6)	4347/4907 (88.6)	0.006
**Age at first admission**				
<50 years	1507/1781 (84.6)	875/1127 (77.6)	868/1034 (83.9)	<0.001
50–69 years	3911/4240 (92.2)	2336/2638 (88.6)	2250/2498 (90.1)	<0.001
≥70 years	1860/2085 (89.2)	1223/1353 (90.4)	1229/1375 (89.4)	0.278
**Type of endocrine therapy**				
Tamoxifen	1975/2210 (89.4)	1075/1278 (84.1)	959/1130 (84.9)	0.001
Aromatase inhibitors	4911/5414 (90.7)	3155/3556 (88.7)	3165/3485 (90.8)	0.001
Combined treatment	392/482 (81.3)	204/284 (71.8)	223/292 (76.4)	0.012

* Adherence defined as 80% coverage of prescribed dose. ^†^ Pre-pandemic: 1st year of treatment completed between January 2017 and February 2020; pandemic: ≥1 dispensation from March to September 2020; post-pandemic: 1st year of treatment completed between October 2020 and December 2022. ^‡^
*p* value.

**Table 3 cancers-16-00426-t003:** Univariable and multivariable analyses of factors associated with adherence to oral endocrine therapy.

	Univariable			Multivariable (*)		
	OR	95% CI	*p*	OR	95% CI	*p*
**Year of first drug dispensation**						
2017	1			1		
2018	0.96	0.82–1.12	0.61	0.95	0.82–1.11	0.54
2019	0.72	0.63–0.84	<0.001	0.72	0.62–0.83	<0.001
2020	0.70	0.60–0.81	<0.001	0.70	0.60–0.81	<0.001
2021	0.89	0.76–1.03	0.11	0.87	0.75–1.01	0.071
**Age group**						
<50 years	1			1		
50–69 years	2.06	1.85–2.29	<0.001	1.87	1.66–2.10	<0.001
≥70 years	1.83	1.62–2.07	<0.001	1.62	1.41–1.87	<0.001
**Type of treatment**						
Tamoxifen	1			1		
Aromatase inhibitors	1.39	1.26–1.55	<0.001	1.11	0.98–1.25	0.096
Combined treatment	0.52	0.44–0.62	<0.001	0.49	0.42–0.59	<0.001

(*) The multivariate model was adjusted for year, age, and type of treatment.

## Data Availability

The datasets generated during and analyzed during the current study are not publicly available due to confidentiality measures but are available from the corresponding author on reasonable request.

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
