# Peer review of "The Impact of the COVID-19 Pandemic on Adherence to Endocrine Therapy for Breast Cancer in Catalonia (Spain)"

_cancers, 2024, doi:10.3390/cancers16020426_

Round 1

Reviewer 1 Report

Comments and Suggestions for Authors

This article is a retrospective cohort study in patients starting endocrine therapy from 2017 to 2021.

It shows the impact of the COVID-19 pandemic on adherence to oral endocrine therapy in patients diagnosed with breast cancer in the public healthcare system in Catalonia (Spain).

The study was approved by the ethics committee for drug research at Hospital Universitari de Bellvitge (reference EOM028/22).

It is an article of importance in its field; the study is original; Introduction adn background informations are complete; overall presentation is adequate and linear.

English language is of sufficient quality and the quality of the figures and tables is satisfactory.

Statistical methods are valid and correctly applied and the methods are sufficiently documented to allow replication studies 

Accuracy of literature citations is good. 

The discussion should be revised and improved. 

The manuscript may be considered eligible to publication with minor revisions.

Comments on the Quality of English Language

English language is  sufficient 

Author Response

Thank you for your comments on our paper. We have introduced some changes in the methods and the discussion following the comments made by the reviewers. 

Reviewer 2 Report

Comments and Suggestions for Authors

The results of this study are not surprising and do not seem very relevant to clinical practice, which limits the interest in this manuscript. However, the study seems to have been conducted correctly.

Some clarifications regarding the objective and methods are necessary and my comments and suggestions for improving the manuscript are as follows:

The COVID-19 pandemic affected the healthcare system, namely, as the authors refer in the Introduction section, breast cancer screening was suspended during March to June 2020 in Catalonia, affecting the screening, diagnosis and cancer stage at diagnosis, and face-to-face consultation and hospitalizations were also reduced in this period. However, as less is known on the impact of the COVID-19 pandemic on breast cancer treatments, the authors focused their study on oral endocrine treatment for breast cancer. Some background information is needed to better understand why the authors hypothesize that adherence to oral endocrine treatment could have been affected by the pandemic. There are different models of dispensation of drugs used in oral endocrine therapy for breast cancer, namely, the dispensation at the hospital pharmacy or at community pharmacies. What model is used in Catalonia? How is the prescription renewed? Are disruptions in the access to endocrine medications expected to have occurred?

As the authors also present and discuss results regarding the use of neo-adjuvant endocrine therapy, maybe the study’s objective should be redefined.

Adherence to drug therapy includes three phases – initiation, implementation, and discontinuation – with regard to a prescribed treatment, that is, if the prescription was made on April, 26, 2018, it is expected that the dispensation and consumption of the drug should start on April, 26, 2018 or in the next day. A longer delay in initiating the treatment should be considered non-adherence. When the authors decide to consider that endocrine treatment begins at the date of the first dispensation, they cannot include non-adherence cases due to delay in initiating the therapy. If prescription date is available in the pharmacy database, it could be used to identify those cases. At least, this limitation should be discussed.

The description of the pre- and post-pandemic categories of treatments should be clarified. I suggest “Pre-pandemic: 1st year of treatment completely performed between January 2017 to February 2020; …; post-pandemic: 1st year of treatment completely performed between October 2020 to December 2022”. Otherwise, treatments that are completed between January 2017 and February 2020 include treatments initiated in February, 1, 2016 and completed in January, 31, 2017, and treatments that are completed between October 2020 and December 2022 include treatments initiated in November, 1, 2019 and completed in October, 31, 2020.

Besides the information in the main text, the authors should add in Table 3 (in footnotes), which variables were included in the multivariate logistic regression model.

Oral endocrine therapy is recommended for women presenting with metastatic disease at least 12 months after completion of adjuvant endocrine therapy or those presenting with de novo metastatic breast cancer. Were these endocrine treatments included in this study? How was age collected in these cases? The statements in lines 72-73 (“This retrospective cohort study included 18,131 patients with breast cancer who initiated oral endocrine therapy from 2017 to 2021, with follow-up to December 2022.“) and in lines 83-87 (“The study cohort was linked to the hospital discharge minimum basic data set in Catalonia to identify women who had been admitted to a public hospital with a primary diagnosis of breast cancer (International Classification of Diseases, 10th revision [ICD-10]: 85 C50, D05.1, D05.7 and D05.9). For each woman in this study population (n = 18,131), age at first admission was recorded.”) are insufficient to clarify if endocrine therapy for recurrent metastatic disease was included in data analysis and what age was considered.

Is it age at first dispensation (Table 2 and lines 101-103) or age at first hospital admission (lines 86-87)?

The authors refer that date of death was collected but do not state how or what for it was used.

How was the date of surgery collected? From the hospital discharge minimum basic data set”? Are data available regarding chemotherapy and radiotherapy? More or less treatments performed could be a factor influencing adherence to endocrine therapy. Could this information be included in the data analysis?

Please clarify the sentence in lines 112-114 “From 2017 to 2021, 21,540 women were diagnosed with breast cancer in the public healthcare system and treated with oral endocrine therapy; of these, 18,131 had adherence data for a full year of follow-up.” as all women who are treated with endocrine therapy between 2017 and 2021 have one year of follow-up, since follow-up ended in December 2022, unless they die first. Do the authors mean that from 2017 to 2021, 21540 women were identified as being treated with oral endocrine therapy and diagnosed with breast cancer, of whom, 18131 had the first dispensation of endocrine drugs after December 2016, and were alive during at least one year after the first dispensation of oral endocrine therapy, allowing for the follow-up of the first year of endocrine treatment of 18131 patients?

Why is no P value presented for the overall non-adherence in Table 1 and Table 2? Crude OR in Table 3 allow for the comparison of adherence between the different civil years but not for the comparison regarding the pre-post, and pandemic periods. I calculated the 95% confidence intervals of each adherence value for the pre- and post-pandemic and the pandemic periods. There were no statistical differences for the pre- and post-pandemic periods but there were statistical differences for the pandemic versus pre- and post-pandemic periods.

The sentence in lines 214-216 “In women under 70 years of age, the recovery in 2021 and 2022 was very notable and statistically significant, while in older women, there was no noticeable impact at all.” should be revised as there are no data for 2022 in the manuscript.

In the sentence in lines 217-218 “The greatest impact was observed in the number of patients starting oral endocrine therapy in 2020 first, reflecting a marked reduction in diagnoses.”, the authors should clarify what impact they are discussing.

Comments on the Quality of English Language

Author Response

We would like to thank the reviewer for the careful and detailed comments made on our paper, which could contribute to its improvement.

We have included background information on the oral endocrine therapy (prescribed by hospital specialist but dispensed at community pharmacy); we added the comment on the absence of disruption of provision of these drugs during the pandemia. 

We have included a sentence in the limitations about the lack of capacity to assess the patients who did not begin the treatment refilling at least one dose of the endocrine therapy.  

We have clarified the definition of the periods of study adding a description of each period as pre or post-pandemic. We hope now is better described. 

We added in a footnote in table 3 the variables in the multivariate analysis.

As it was mentioned in the discussion (limitations paragraph), we do not have the stage at diagnosis from the clinical records as it is a variable not structured in our region. Then, treatment for metastatic patients is included. Age was collected using the same approach, using the date of the hospitalization for the ifirst time with a diagnosis of breast cancer. 

Age was included as age at first hospital admission in all cases with a diagnosis of breast cancer. We have now clarified this point.

Date of death in this study was only collected for the purpose of identifying the date of last day of therapy within the first year of treatment. We have added a short sentence to clarify this point.

Data of surgery was collected from the hospital discharge data set. It is now better described. 

Regarding the number of cases in figure 1, the 21540 women were identified with a primary diagnosis of breast cancer between 2017 to 2022 but we excluded those diagnosed in 2022 because they could not have a follow up of one year because they do not have one full year of treatment completed. 

All these points are included as tracked changes in one of the manuscript included.

Reviewer 3 Report

Comments and Suggestions for Authors

The reviewed article concerns an important issue: the impact of the COVID-19 pandemic on the effectiveness of therapy for patients with breast cancer. Understanding the causes of problems with continuing therapy is important - it can help identify risk factors. This is especially important in cancer patients, where the risk of death is high. The undoubted strength of the article is the large group of respondents, which makes the results of the obtained analyzes reliable.

The article has the correct structure. The statistical analysis methods used are correct and well described. The results are presented in a clear way and are well and clearly described.

I have a few comments: The introductory part lacks an analysis of the causes of problems related to adherence to therapy for breast cancer (as well as in the case of other diseases). Are these factors psychosocial, demographic or perhaps related to socioeconomic status? Such a short analysis of the conditions would help to better understand the described phenomenon.   The second comment is related to the first. Very few factors were taken into account in the analysis. A large group would allow for the creation of more complex regression equations. There is a significant lack of factors present in many methodologically similar studies, such as the education of the patients.        

Author Response

Thanks to the reviewer for the comments and suggestions made and for the global evaluation of the paper.

Following your suggestions, we have included a sentence in the discussion to remark the limited number of factors associated with adherence analysed in this paper due to the approach based on real-world data. There is a specific reference (31) that reviews all the factors found associated with adherence. Our approach to the introduction was very focused.